# Current Status of Radiolabeled Monoclonal Antibodies Targeting PSMA for Imaging and Therapy

**DOI:** 10.3390/cancers15184537

**Published:** 2023-09-13

**Authors:** Mohammed Abusalem, Lucia Martiniova, Sarita Soebianto, Louis DePalatis, Gregory Ravizzini

**Affiliations:** 1Department of Nuclear Medicine, The University of Texas MD Anderson Cancer Center, Houston, TX 77030, USA; 2Department of Experimental Therapeutics, University of Texas MD Anderson Cancer Center, Houston, TX 77030, USA; 3BioDevelopment Solutions, LLC, 226 Becker Circle, Johnstown, CO 80534, USA

**Keywords:** PSMA, radiolabeled mAbs, prostate cancer, imaging, targeted radionuclide therapy

## Abstract

**Simple Summary:**

The prostate-specific membrane antigen (PSMA) protein present in most prostate cancer cells has emerged as a promising target for the imaging and treatment of prostate cancer. With the development of monoclonal antibodies (mAbs) against PSMA, cancer cells can be targeted effectively while minimizing side effects to patients. Radioactivity can be attached to mAbs for imaging and treatment. This manuscript provides an overview of the current development and future prospects of radioactive mAbs conjugates targeting PSMA in prostate cancer. The current FDA approved PSMA radioligand therapy (Pluvicto^TM^) faces some barriers in the form of frequent side-effects such as dry mouth. As a result, prostate cancer treatment based on radioactive mAbs conjugates hopes to reduce side-effects while still providing effective treatment to patients with prostate cancer.

**Abstract:**

Prostate cancer (PCa) is one of the most prevalent cancer diagnoses among men in the United States and in several other developed countries. The prostate specific membrane antigen (PSMA) has been recognized as a promising molecular target in PCa, which has led to the development of specific radionuclide-based tracers for imaging and radiopharmaceuticals for PSMA targeted therapy. These compounds range from small molecule ligands to monoclonal antibodies (mAbs). Monoclonal antibodies play a crucial role in targeting cancer cell-specific antigens with a high degree of specificity while minimizing side effects to normal cells. The same mAb can often be labeled in different ways, such as with radionuclides suitable for imaging with Positron Emission Tomography (β+ positrons), Gamma Camera Scintigraphy (γ photons), or radiotherapy (β− electrons, α-emitters, or Auger electrons). Accordingly, the use of radionuclide-based PSMA-targeting compounds in molecular imaging and therapeutic applications has significantly grown in recent years. In this article, we will highlight the latest developments and prospects of radiolabeled mAbs that target PSMA for the detection and treatment of prostate cancer.

## 1. Introduction

Researchers continue to develop new antibody-based diagnostics and therapeutics for cancer based on their demonstrated effectiveness and ability to selectively bind to specific receptors (or antigens) that are overexpressed in tumor cells. While non-antibody ligands, such as transferrin or folate, may be useful targeting agents for their specific receptors on tumor cells, the same ligands are also abundant in certain normal cells, thereby leading to unacceptably low tumor-to-normal tissue ratios and consequent cytotoxicity. When it comes to specificity and selectivity, mAbs serve as much better vectors for the precise delivery of imaging or radiotherapy components to tumor cells while sparing healthy normal cells [1]. Furthermore, mAbs can be conjugated with a number of radionuclides that are chosen based on desirable half-lives and target tumor characteristics such as size, morphology, and location [2]. A disadvantage of mAbs is that they have the potential to elicit an immunogenic response in humans, especially if they are not of human origin. Generally, these responses may manifest as hypersensitivity reactions such as anaphylactic shock and decreased therapeutic response to subsequent treatments with the same antibody targeting agent [3,4].

The tumor uptake of mAbs is affected by several factors, including blood flow, vascular volume, and antibody size. Additionally, the blood clearance of mAbs is slow and liver uptake is pronounced [5]. By using antibody fragments rather than whole antibodies, immunogenicity and serum half-life can be reduced [3]. However, in order to achieve a sufficient tumor-to-background ratio, one might consider using a targeting moiety with a longer half-life if the expression of a particular target is relatively low. With radiolabeled mAbs, personalized imaging can be used to select patients who may benefit from subsequent mAb treatment [6].

The prostate-specific membrane antigen (PSMA) is a zinc containing metalloenzyme having a molecular weight of 100 kDa and is overexpressed in the vast majority of prostate cancer cells and in tumor neovasculature [7,8]. Figure 1 describes the symmetric dimer with each polypeptide chain containing a protease domain, an apical domain, and a helical domain. At the periphery of these domains is a large cavity containing a binuclear zinc binding site [9].

As a glutamate-carboxypeptidase, PSMA functions as a folate hydrolase to catalyze the hydrolysis of N-acetyl aspartyl glutamate (NAAG) and as an NAALADase (N-Acetylated alpha-linked acidic dipeptidase), which is a neuropeptidase that regulates glutamatergic neurotransmission [10]. These enzymatic processes are relevant for the detection of prostate cancer, as it was shown that one type of NAALADase is over-expressed in prostatic adenocarcinomas [11]. Additionally, high folate hydrolase activity has been reported in human prostate cancer cells producing PSMA [12]. Also, recent studies have demonstrated that PSMA elicits a signal that allows the protein on the cell surface to be internalized into the endosome [13]. The five N-terminal amino acids (MWNLL) located at the cytoplasmic tail of PSMA facilitate its internalization, which occurs through a clathrin-mediated endocytic mechanism as demonstrated by Rajasekaran et al. [14]. This important feature is advantageous when considering PSMA as a promising antigenic biomarker for imaging or therapy of prostate cancer [13]. PSMA expression is up to 1000-fold higher in secretory cells of prostate cancer epithelium, and is positively associated with high Gleason scores, high serum prostate-specific antigens (PSAs), advanced tumor stages, and a high likelihood of recurrence of the disease [15,16]. Additionally, PSMA tends to be expressed in inverse proportion to androgen concentration and increases when cancer cells become independent of androgen [17]. In total, the aforementioned findings make PSMA a promising and effective diagnostic and prognostic biomarker of prostate cancer [18].

Currently, prostate cancer is one of the most prevalent cancer diagnoses among men in the United States and in a number of other countries [19,20]. It is expected that around 288,300 new cases will be diagnosed in 2023 in the United states alone [21]. In nuclear medicine practice, PSMA has been targeted using two different approaches. First, radiolabeled enzyme inhibitors or binding agents are used to identify targets based on the enzymatic activity of PSMA [22]. The second approach utilizes the macromolecular structure of PSMA as a targeting vector for small molecule radioligands or monoclonal antibodies [23].

In the first approach, radiolabeled small molecules are used to target PSMA enzyme activity on the basis that blood clearance rates for smaller molecules are more rapid than for whole mAbs. The result is a higher target-to-background ratio. Furthermore, radiolabeled small molecules are internalized via clathrin-coated pits followed by endocytosis after binding to their membrane-bound targets [7]. Over the last decade, the development of radiolabeled peptide targeting technologies has focused on developing small molecule ligands that inhibit the binding of the NAAG substrate to the extracellular portion of PSMA [7]. Currently, there are several Food and Drug Administration (FDA) approved PSMA small molecule radioligands, including: ^68^Ga-PSMA-11, ^18^F-DCFPyL, and ^18^F-rhPSMA-7.3 [24,25] for diagnostic imaging, and ^177^Lu-PSMA-617 (Pluvicto^TM^) for radiotherapy. [26] Interestingly, these and other PSMA ligands have different kinetic profiles. ^68^Ga-PSMA-11, ^18^F-DCFBC, ^18^F-DCFPyL, and ^177^Lu-PSMA-617 are urea-based radioligands that contain Lys-ureido-Glu pharmacophores to target PSMA and have established diagnostic and therapeutic capabilities in humans [27,28,29]. A more recent addition to PSMA targeting is a radiohybrid PSMA (rhPSMA). This class of radiopharmaceuticals can be labeled either with ^18^F or with radiometals. rhPSMA ligands have a number of unique properties that can be pertinent to imaging and theranostic applications due to their reduced urinary excretion in comparison to older generation PSMA radioligands [30,31]. However, patients imaged with F-18-rhPSMA-7.3 still show frequent significant urinary activity in the urinary bladder that may impair its diagnostic utility, particularly in the evaluation of tumors in the prostate gland or prostatectomy bed. Improvements were made with the synthesis of ^18^F-PSMA-1007, a radioligand that is structurally related to and has pharmacologic properties similar to those of PSMA-617. The optimal imaging time for this compound is at 120 min post injection and is excreted primarily via the hepatobiliary system [32]. Another PSMA inhibitor that is rapidly excreted through the kidneys is PSMA I&T. This targeting agent was labeled with ^68^Ga and ^64^Cu (ClinicalTrials.gov Identifier: NCT05653856) for imaging and ^177^Lu for therapy [33]. Unfortunately, both PSMA-617 and PSMA I&T pose increased radiation exposure to the kidneys due to rapid urinary excretion [34]. Therefore, a new approach of introducing albumin-binding groups into small molecules has been proposed as a way to increase their blood circulation time and enhance the tumor uptake of otherwise rapidly cleared molecules, while reducing non-target tissue doses due to reduced tracer uptake [35,36]. As an example, we note the results from a recent Phase I trial with Evans-Blue-modified PSMA-617 derivative (^177^Lu-EB-PSMA-617) [37,38]. As reported by the authors, EB-PSMA-617 retained a high degree of internalization by tumor cells, and a prolonged time window for binding to PSMA, thus significantly increasing the tumor accumulation of the targeting agent [39].

The second approach, where PSMA has been used as a target, utilizes the macromolecular structure of PSMA to provide specific monoclonal antibodies for use as targeting vectors. Specific examples of these mAb compounds will be discussed in the following section. Therefore, the aim of this review is to provide an overview of the status of anti-PSMA radiolabeled mAbs for the imaging and therapy of prostate cancer collected from peer-reviewed articles published through July 2023.

## 2. Radiolabeled Monoclonal Antibodies Targeting PSMA

In general, it is best to match the biological half-life of mAbs with the half-life of the radionuclide when developing radiolabeled diagnostic and therapeutic agents. Several important differences exist between small molecule radioligands and radiolabeled monoclonal antibodies, including their binding sites on PSMA, their pharmacokinetics (PK), and biodistribution properties [40]. An intact IgG mAb has a molecular weight of approximately 150 kDa. This large molecular weight prevents clearance by glomerular filtration and results in a relatively long biological half-life of 3–7 days or longer. Small molecules such as affinity proteins or peptides are excreted rapidly through the kidneys (in vivo half-life of approximately 3 h) [40]. Due to the smaller size of mAb fragments, they possess faster PKs and clearance rates, and can therefore be labeled with short half-life radionuclides, e.g., ^18^F, ^64^Cu, and ^68^Ga, for effective imaging applications [41].

PSMA-targeted radionuclide therapeutic agents contain either small molecules or antibodies that are conjugated to radionuclides that emit α or β particles in order to deliver a sufficiently large radiation dose to the tumor [42]. This section will describe key radiolabeled mAbs targeting PSMA which are currently under development or in clinical trials (Figure 2).

### 2.1. ^111^In-7E11-C5

In 1996, the U.S. FDA approved Indium-111 (^111^In)-capromab pendetide (Prostascint^®^, Cytogen Corporation, Princeton, NJ, USA), the first anti-PSMA radiolabeled monoclonal antibody for gamma-scintigraphy in patients with prostate cancer. (^111^In)-capromab pendetide is a conjugated murine IgG1 mAb (7 × 10^11^) that recognizes the PSMA epitope localized in the intracellular (cytoplasmic) domain of PSMA (Figure 2) [43,44]. ^111^In-capromab pendetide offered improved detection when compared to conventional imaging for initial staging, with a sensitivity of 52–62% and a specificity of 72–96% [45]. In the setting of biochemical recurrence after radical prostatectomy, radiation therapy, and/or hormonal therapy, Elgamal et al., reported a sensitivity of 89%, a specificity of 67%, and an overall accuracy of 89% [46]. However, the patient population included in this study had a high average serum PSA concentration of 55.9 ng/mL at the time of the scan. (^111^In)-capromab pendetide localized primarily to the liver, spleen, and bone marrow [5]. (^111^In)-capromab pendetide scan was considered a valuable tool for the staging of prostate cancer with the predictive ability superior to that of CT and MRI in detecting lymph node metastases [47]. The fundamental drawback of ^111^In-capromab pendetide is that the antibody is thought to bind to dead or dying cells but fails to recognize viable cancer cells [44]. Clinically, this agent did not perform well due to poor pharmacokinetics and the inability to reach its target epitope on PSMA [48]. Figure 3D shows an example of a (^111^In)-capromab pendetide single photon emission computed tomography (SPECT) scan with positive findings of two foci of increasing radiotracer activity in the right and left prostate beds. SPECT/CT delayed images of a (^111^In)-capromab pendetide scan are usually acquired at 30 min and between 72–120 h post-injection due to the elevated blood pool activity as presented on the early phase images (Figure 3A,B). Normal physiologic uptake includes prostate tissue (Figure 3C) and non-androgen-dependent sites including salivary epithelial tissue, liver, spleen, bone marrow, and bowel with urinary excretion [49].

### 2.2. ^89^Zr-J591

More recently, second and third generation PSMA-targeting mAbs were designed to overcome the limitation of capromab pendetide. This led to the development of humanized mAbs (huJ591), which targeted the extracellular domain of PSMA (Figure 2) [51,52]. Initially, the bifunctional chelating agent, DOTA, was conjugated to huJ591 and labeled with ^111^In for single photon gamma camera imaging [53]. Subsequently, radiolabeled J591 has been clinically evaluated for PSMA Positron Emission Tomography (PET) imaging with Zirconium-89 (^89^Zr) and for therapy applications using Yittrium-90 (^90^Y) or Lutetium-177 (^177^Lu) [54,55]. Figure 4 depicts the biodistribution of ^89^Zr-J59. The radiotracer activity is seen mainly in the liver, spleen, and kidneys. In one of the first reported feasibility studies using ^89^Zr-J591 for PSMA PET/CT imaging, ^89^Zr-J591 was able to identify larger tumors in 11 patients with localized PCa before undergoing radical prostatectomy; however, there was no difference in Gleason scores between ^89^Zr-J591 positive and ^89^Zr-J591 negative tumors [54]. In subsequent clinical trials (Trial registration ID: NCT01543659), ^89^Zr-J591 PSMA PET imaging demonstrated detection of metastases in patients with progressive metastatic castration-resistant prostate cancers (mCRPC). Overall accuracy was reported at 95.2% and 60% for osseous and soft tissue lesions, respectively, with an optimal post-injection imaging time at 6–7 days [56,57]. Cornelis et al. [58] evaluated ^89^Zr-J591 for PET guided biopsy and found it to be highly accurate for prostate cancer. With the advantage of a longer half-life compared to ^18^F-FDG, ^89^Zr can help avoid reinjection for biopsy guidance [58]. To reduce the imaging time, (due to the slow clearance of the J591 antibody), a minibody that was subsequently labeled with ^89^Zr was produced (Df-IAB2M; MW: ~80 kDA). The minibody was generated without the Fc-receptor domain via genetic engineering to allow faster blood clearance, thereby shortening the imaging time to 48 h post injection [59].

J591 showed excellent targeting and imaging capabilities for PSMA. However, its translation to the clinic has been hampered for a number of reasons including its poor tumor penetrability, extended delays between agent administration and imaging, and its slow clearance from the blood pool [60]. Nevertheless, the therapeutic efficacy of J591 has been shown to be effective and produced desirable patient outcomes when targeting mCRPC, as demonstrated by clinical trials data (Figure 4) [61].

**Figure 4 cancers-15-04537-f004:**
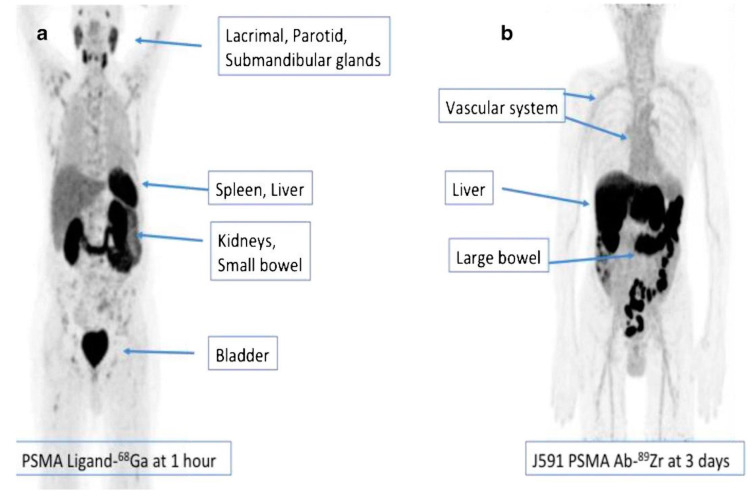
Biodistribution of two different PET imaging agents showing different patterns of distribution for ^68^Ga-PSMA-11 (**a**) 1 h after injection of radiotracer and Zirconium-89 labeled J591 antibody and (**b**) 3 days after radiotracer injection. With permission adapted from Sun et al. [61].

### 2.3. ^177^Lu-J591

Lutetium-177 (^177^Lu) is an ideal isotope for combined radioimmunotherapeutic and radiodiagnostic applications. It has an acceptable physical half-life of 6.7 days and emits short-range (0.2–0.3 mm) β particles and auger electrons for therapy as well as γ emission used for diagnostic SPECT imaging and dosimetry [62]. Initially, ^177^Lu-J591 was used in a clinical phase I study in 2005 where it was shown to target PCa metastatic sites in 35 patients. Multiple doses of 30 mCi/m^2^ were well tolerated, with a maximum tolerated dose (MTD) of 70 mCi/m^2^ [63]. Moreover, in another Phase II study, 10.6% of patients (total of 47) had a decline of ≥50% in the level of PSA, while 36.2% of the patients experienced a ≥30% PSA decline after receiving treatment with ^177^Lu-J591 (clinicaltrials.gov NCT00195039) [55]. Tagawa et al. [64] proposed a dose fractionation regimen since it may allow higher doses to be administered safely. In their Phase I/II study (NCT00538668), ^177^Lu-J591 was evaluated in patients with castration resistant prostate cancer. As the dose was increased from 20 to 45 mCi/m^2^, increased hematologic toxicity and therapeutic efficacy were reported. In 29.4% of patients receiving two doses given two weeks apart at 45 mCi/m^2^, a more than 50% PSA decline with median overall survival of 42.3 months was confirmed [64]. As previously reported, the most common toxicity was myelosuppression, which was reversible (Figure 5). Additionally, the authors discussed the option of fractionated administration of ^177^Lu-J591 to increase the cumulative radiation dose. PSA decreased, while overall survival and toxicity (dose-limiting myelosuppression) increased with higher doses [64]. Additionally, Bander et al. [63] evaluated organ dosimetry and reported that the liver was the critical organ with the highest radiation dose (~7 cGy/mCi), followed by the spleen and kidneys. No significant uptake or non-detectable levels in salivary glands were reported, as confirmed by ^89^Zr-J591 PSMA PET imaging [40].

### 2.4. ^117^Lu-DOTA-TLX591

At present, a Phase I clinical trial (NCT04786847) sponsored by Telix Pharmaceuticals is underway to evaluate the safety, tolerability, biodistribution, and dosimetry of ^177^Lu-DOTA-TLX591 administered to PSMA+ mCRPC patients together with the best standard of care (SoC). Patients enrolled in this trial had progressive disease, despite prior treatment with a novel androgen axis drug (NAAD) [65]. Patients will be recruited for evaluation of biodistribution of TLX591 administered in combination with SoC. These patients will receive a single tracer (27 mCi) intravenous (IV) infusion of TLX591, and SPECT images and pharmacokinetic blood samples are acquired at several time points [65]. The study is expected to be completed in late 2024. ^177^Lu-DOTA-rosopatamab (also referenced to as TLX591 antibody sponsored by Telix Pharmaceuticals) is currently undergoing evaluation in a multicenter Phase III clinical trial (NCT04876651). This trial is actively recruiting patients with mCRPC that express PSMA and have progressed despite prior treatment with a NAAD. This clinical trial is designed to investigate and confirm the benefits and risks associated with the PSMA-targeted antibody, ^177^Lu-DOTA-rosopatamab, administered together with SoC, as compared to the best SoC alone [66]. Approximately 387 patients will be randomized to one of two groups in a 2:1 ratio to receive one of the following treatments; Group A: Two single intravenous (IV) injections of 76 mCi each (equivalent to a 45 mCi/m^2^ dose in a standard 1.7m^2^ individual) of ^177^Lu-DOTA-rosopatamab, given 14 days apart, plus best SoC, and Group B: Best SoC [66]. The estimated completion date of the study is in 2027. Thus far it has been reported that TLX591 is effective at targeting tumor-expressed PSMA and is cleared by the liver [67].

### 2.5. ^64^Cu-TLX592

Using Telix’s proprietary RADmAb^®^ antibody technology, TLX592 has been designed to clear a patient’s circulation much more rapidly than unmodified antibodies. ^64^Cu-TLX592 consists of a humanized, engineered monoclonal antibody, HuX592r, conjugated with a DOTA chelator and radiolabeled with ^64^Cu for PET imaging. Safety, PK, Biodistribution, and Dosimetry of ^64^Cu-TLX592 was studied in a Phase I clinical trial, CUPID (ClinicalTrials.gov Identifier: NCT04726033). As mentioned for TLX591, TLX592 also maintains its hepatic clearance, and retains the high affinity to tumor PSMA. Additionally, the rapid clearance from the circulation of patients suggests that TLX592 may be a more appropriate targeting agent for alpha particle-based therapies [67]. Therefore, the results of the CUPID study will evaluate the feasibility of ^64^Cu-TLX592 PET imaging prior to initiation of targeted alpha therapy with ^225^AC-TLX592 [67].

### 2.6. ^225^Ac-J591

Radiolabeling of J591 mAb with alpha emitting payloads was investigated after evidence of a therapeutic dose response following single treatment or fractionated dosing of ^177^Lu-J591 mAb [55,64]. Tagawa et al. [68] reported results of a dose-escalation first-in-human study of ^225^Ac-J591 with an expansion cohort of 32 patients with progressive mCRPC (NCT03276572). Dose-escalation was in single participants × 4 followed by 3 + 3 with a single infusion of ^225^Ac-J591 (13.3 KBq/kg with planned escalation up to 93.3 KBq/kg) [68]. Dose limiting toxicity (DLT) was seen in 1 of 6 participants in cohort 6 (80 KBq/kg) with DLT grade 4 anemia and thrombocytopenia. High-grade adverse effects were restricted to hematologic parameters. In addition to DLT, four participants (12.5%) developed grade 3 thrombocytopenia and two (6.2%) developed grade 3 neutropenia. Additionally, adverse effects included grade 1 xerostomia (of which five received prior ^177^Lu-PSMA) and twelve (37.5%) with AST elevation [68]. Almost half (37.5%) of the cohort experienced a PSA decline greater than 50%. Four patients (all from the 80 KBq/kg cohort) continued to be progression-free and on study after more than six months [69]. ^225^Ac-J591 is currently in a Phase I/II dose escalation trial (NCT04506567) to determine the highest dose level of ^225^Ac-J591 that can be given without severe side effects when targeting progressive mCRPC [70].

Clinical studies suggest that the alpha emitter ^225^Ac may be an ideal radionuclide for radiotherapy, as it offers the promise of delivering greater energy (i.e., higher biological killing efficacy) over shorter distances and less bystander effects on normal tissue. Kratochwil et al. [71] demonstrated the radiotherapeutic efficacy of ^225^Ac in the first in-human treatment when labeled with PSMA-617. They reported a complete therapeutic response after semi-monthly administration of 100-kBq of ^225^Ac-PSMA-617/kg body weight, as shown by declining PSA levels upon imaging [71].

Bander et al. reported on preliminary trial (NCT04946370) results combining ^225^Ac-J591 and Pembrolizumab in mCRPC [72,73]. In the phase I clinical trial, patients received androgen receptor pathway inhibitor (ARPI) and pembrolizumab (immunotherapy) plus a single infusion of ^225^Ac-J591 at 65 or 80 KBq/kg. The primary endpoint for phase I was the determination of the recommended phase II dose of ^225^Ac-J591. This ongoing phase I/II study will assess whether ^225^Ac-J591 + pembrolizumab + ARPI is more effective against prostate cancer than pembrolizumab + ARPI alone [73]. Phase I preliminary results indicated that PSA levels decreased in all twelve patients receiving the regimen, with six patients in the higher-dose cohort experiencing declines greater than 50%. A transient syndrome occurred unexpectedly in seven patients between 7 and 14 days post-treatment. The patients experienced fever over 101 ^•^F, plus low platelet and leukocyte counts, as well as a morbilliform rash, all common side effects associated with immune checkpoint inhibitors [72].

### 2.7. Comparison of Biodistribution of ^177^Lu-PSMA-617, ^177^Lu-J591, and ^225^Ac-J591

The major differences between the small molecule radioligand PSMA-617 and the radiolabeled monoclonal antibody J591 are in their size, biodistribution properties, and blood circulation. Whole mAbs such as J591 are larger molecules (~molecular weight of 150 k) with a long circulation time and blood clearance, which makes the optimal imaging time for tumors several days after injection (Figure 5) [74]. As a consequence of larger molecular weight and slow blood clearance, hematologic toxicity is a frequent occurrence in patients. As reported by Tagawa et al., grade 4 thrombocytopenia and grade 4 neutropenia occurred in 46.8% and 25.5% of patients, respectively [55]. In contrast to antibody-based treatments, small molecules targeting PSMA result in reduced bone marrow toxicity. The PSMA-617 ligand, with a shorter circulation time and lower molecular weight, has favorable pharmacokinetic and tumor penetration properties [74]. Pharmacokinetics and biodistribution of ^177^Lu-J591 was described by Vallabhajosula et al. [53]. They reported that the liver received a high radiation dose, followed by the spleen and kidney, making the liver the critical organ. Organs that receive the highest absorbed doses of ^177^Lu-PSMA-617 are the lacrimal glands, salivary glands, large intestine, kidneys, rectum, and urinary bladder wall [75]. Notably, the low uptake of ^177^Lu-J591 uptake in salivary glands presents as a theoretical advantage for ^177^Lu-J591 over ^177^Lu-PSMA-617. When comparing ^177^Lu-J591 with results published from the VISION trial (NCT03511664 phase 3 randomized trial of ^177^Lu-PSMA-617 plus best standard/standard-of-care (SoC) versus best SOC care alone in heavily pretreated mCRPC patients), xerostomia (dry mouth) occurred in 38.8% of patients (grade 3 or higher) in the ^177^Lu-PSMA-617 experimental cohort [76]. On the other hand, only one patient reported xerostomia in the ^177^Lu-J591 trial [64]. Nauseef et al. [77] investigated the MTD of ^225^Ac-J591 in a first-in-human phase I dose-escalation study in mCRPC. They reported that no MTD was reached in patients treated with a single dose of ^225^Ac-J591 at seven different dose cohorts (up to dose 93.3 KBq/kg) [77]. In a parallel dose-escalation study, single cycle and fractionated multiple dose regimens (q6w) were used in 24 predominantly ^177^Lu-naïve patients [78]. A single fractionated dose of ^225^Ac-J591 resulted in dose-limiting toxicity (DLT); namely, neutropenia (grade 4 or febrile neutropenia) and thrombocytopenia (grade 4 or grade 3 with clinically significant hemorrhage). The most common low grade non-hematologic treatment adverse effects were fatigue (95%), xerostomia (69%), and nausea (57%) [78].

### 2.8. ^227^Th-PSMA-TTC

Hammer et al. [79] described a novel PSMA-targeted thorium-227 antibody-based conjugate, PSMA-TTC (BAY 2315497), in animal models. It consists of the α-particle-emitting thorium-227 (^227^Th, 18.7 days half-life) complexed with a 3,2-HOPO chelator covalently linked to a fully human PSMA-targeting antibody. In preclinical evaluation studies, PSMA-TTC demonstrated strong antitumor activity. Clinical trials were initiated for PSMA-TTC in a Phase I dose-escalation study (NCT03724747) to determine the safety, tolerability profile, and MTD of PSMA-TTC in mCRPC patients. A second objective was to specify the recommended dose of PSMA-TTC for future clinical trials after assessment of the biodistribution and clearance of the drug from the body [80].

### 2.9. ^225^Ac-PSMA-TAC

Another potential antibody targeting PSMA is pelgifatamab. This antibody is covalently linked to a binding moiety of ^225^Ac. According to Schatz et al., preclinical studies revealed improved efficacy and tolerability, faster clearance from normal organs, and enhanced antitumor efficacy [81]. Clinical studies of ^225^Ac-pelgifatamab (^225^Ac-PSMA-TAC) in prostate cancer patients are strongly recommended based on these preclinical findings [81]. This PSMA-targeted antibody is in the early stages of development to treat metastatic castration-resistant prostate cancer. A license for PSMA-TAC has been granted to Bayer Pharmaceuticals. Currently, ^225^Ac-PSMA-TAC is undergoing investigation in a Phase II clinical trial (ClinicalTrials.gov Identifier: NCT05219500) in mCRPC.

## 3. Preclinical Studies and Future Directions

A new prostate cancer cell surface target, CD46, was found and validated using the UA20 single-chain antibody fragment (scFv). The antigen was found to display lineage independent homogeneity in both adenocarcinoma and neuroendocrine prostate cancer [82]. The observations with the UA20 scFv were used to generate a panel of unique scFvs that were ultimately converted to whole human IgG1 antibody forms. One of these, YS5, binds to a tumor selective conformational epitope, enabling therapeutic targeting of CD46. YS5 was characterized and developed for use in multiple clinical trials (ClinicalTrials.gov Identifier: NCT03575819 and NCT05011188) [82,83]. Li et al. [84] described a YS5-based anti-CD46 antibody labeled with ^212^Pb (^212^Pb-TCMC-YS5) in a subcutaneous mCRPC cell line-derived xenograft (CDX), an orthotopically grafted mCRPC CDX model (ortho-CDX), and a prostate cancer patient-derived xenograft model (PDX) [84]. A single dose of 0.74 MBq (20 Ci) of ^212^Pb-TCMC-YS5 was well tolerated in all three models. Furthermore, a lower dose of ^212^Pb-TCMC-YS5 (0.37 MBq) was assessed on the PDX model, and this also demonstrated a significant tumor growth inhibition and prolongation of animal survival [84]. YS5 is also undergoing evaluation as a diagnostic immunoPET agent (NCT05245006) in a first-in-human study of ^89^Zr-DFO-YS5 for detection of CD46 positive malignancy in men with prostate cancer.

Bidkar et al. [85] reported biodistribution and toxicity analysis with [^225^Ac]-DOTA-YS5 for PSMA-positive and PSMA-negative tumors in preclinical PDX and other xenograft models. Biodistribution studies have demonstrated high tumor uptake of [^225^Ac]-DOTA-YS5 over 17 days. The [^225^Ac]-DOTA-YS5 was gradually cleared from all healthy organs except the liver and bones. An evaluation of the therapeutic efficacy of [^225^Ac]-DOTA-YS5 in prostate cancer models is underway. A toxicology evaluation revealed that the 0.5 Ci activity levels were toxic to the kidneys, most likely because ^213^Bi was redistributed [85].

Mazzocco et al. [86] developed an orthotopic model generated by injecting LNCaP cells into the prostate to evaluate a fluorescent-labeled scFv of the anti-PSMA antibody, scFvD2B, as a specific probe for the detection of prostate cancer by in vivo fluorescence imaging. The results of their study demonstrate that scFvD2B is a highly specific imaging agent for the in vivo detection of PSMA-expressing cells in the prostate. Prior studies have demonstrated the specificity and binding properties of scFvD2B and its ability to internalize both in vitro and in vivo in PSMA-expressing cells [87]. Data from this study indicate that intravenously administered NIR fluorescent-labeled scFv targeted the tumor and was internalized by PSMA-expressing cells. The fluorescence signal associated with scFvD2B within tumors was quantified over time. Figure 6 represents a detection of the X770-scFvD2B probe in the prostate at 72 h post-injection. Histochemical and immunohistochemical methods were used to confirm the presence of prostatic carcinoma cells and to detect the X770-scFvD2B probe in the prostate. A control scFv antibody fragment, scFvD2BGF7.7, did not recognize PSMA and did not label the malignant prostate tissue.

## 4. PSMA-Based mAbs vs. Small Molecular Ligands

The sheer volume of antibody-based radiopharmaceuticals under clinical trials is yet another testament to the progress that has been made over the years to provide more effective, safe, and personalized treatments for cancer patients. Indium-111 (^111^In)-capromab pendetide was the first PSMA-targeting imaging probe to receive regulatory approval for the detection of prostate cancer. However, its clinical use was limited since it targeted the internal domain of PSMA. Recently, small molecular ligands targeting PSMA have gained more attention due to their attractive properties such as faster tumor uptake, rapid excretion, and high affinity towards PSMA [88]. At present, there are three small molecule radiotracers approved by the United States FDA for PSMA imaging: Ga-68 gozetotide (^68^Ga-PSMA-11), F-18 piflufolastat (^18^F-DCFPyL PSMA), and F-18 flotufolastat (^18^F-rhPSMA-7.3). The latter was granted FDA marketing approval in May 2023.

Several studies have shown that PSMA theranostics are an effective treatment for prostate cancer. As an example, ^177^Lu-PSMA-617 for mCRPC targeting has fewer side effects [89]. Nevertheless, the number of clinical trials using anti-PSMA mAbs to treat mCRPC in radiopharmaceutical therapy (PRT) applications are showing promising results that could signify a turning point in the treatment of progressive mCRPC. Table 1 summarizes all mentioned clinical trials currently utilizing radiolabeled mAbs in prostate cancer. A major barrier to targeted PSMA radioligand therapy is the presence of significant doses delivered to the salivary and lacrimal glands which may lead to xerostomia and dry eyes [90]. MAbs described in this review have minimal salivary gland uptake. Due to differences in biodistribution and radiation dosimetry, small molecule radioligands may be given in conjunction or sequentially before or after radiolabeled mAbs. Further refinements in radiopharmaceutical development could involve utilizing antibody fragments or minibodies tailored to specific clinical situations. Due to their relatively small size, these constructs will have faster blood clearance and higher affinity to the PSMA, and presumably, higher degrees of internalization compared to whole mAbs. Yokota et al. demonstrated that scFvs may represent a unique targeted delivery system for drugs, toxins, or radionuclides to a tumor site [4]. When compared to whole antibodies, scFvs offer several advantages, including a lower immunogenicity because of their relatively small size and they may also be capable of greater tumor penetrance [4].

While there are a number of possible future directions for PSMA theranostics, we will briefly highlight Cu-based radiopharmaceuticals as representative of where the field may go for other molecules in this general class. Taking mAbs into consideration with their long blood circulations times, diagnostic ^64^Cu has a favorable half-life (12.7 h) and β+ emission (17.4%). Additionally, radioisotopes of Cu include the theranostic pair ^67^Cu (half-life, 61.9 h; β− emitter (100%)) [91,92]. The potential benefits of a Cu radiopharmaceutical will depend on how well it is retained in tumor tissue and how quickly it is cleared from normal tissue. It has been reported that the use of chelators that form Cu complexes that are susceptible to release the metal in vivo can lead to high liver uptake during late-stage treatment [93]. It was reported by Hicks et al. [94] that Cu(II) complexes of sarcophagine (Sar)-based ligands were stable in vivo when conjugated to peptides and antibodies with pendent carboxylate functional groups [95]. Zia et al. [27] described a conjugation of Sar to PSMA at room temperature as an aqueous solution with high radiochemical purity (>97%) and specific tumor binding to LNCaP human prostate adenocarcinoma cells. Figure 7A depicts comparative tumor uptake data ^64^Cu-CuSarPSMA (2 MBq, 0.9 nmol of peptide, monomer) and ^64^Cu-CuSarbisPSMA (2 MBq, 0.2 nmol of peptide, bivalent). The biodistribution of bivalent ^64^Cu-CuSarbisPSMA revealed significantly higher tumor uptake and retention in comparison to the monomer, ^64^Cu-CuSarPSMA (Figure 7A,B), specifically binding to PSMA (Figure 7B) [27]. McInnes et al. [91] investigated the therapeutic efficacy of ^64^Cu-CuSarbisPSMA in LNCaP animal models and compared results obtained to those for ^177^Lu-LuPSMA I&T. The experiments confirmed comparable inhibition of tumor growth for both compounds (Figure 7C). With regards to future directions and clinical applications, the longer half-life of ^64^Cu-Sar-bisPSMA allow for this compound to be used for dosimetry calculations [96]. Clinical trials are ongoing to evaluate the diagnostic and therapeutic performance of ^64^Cu-CuSar-bisPSMA and ^67^Cu-CuSar-bisPSMA (NCT04839367/ NCT04868604/ NCT048393671).

## 5. Conclusions

There are currently a number of small molecule radioligands that have been approved by the United States Food and Drug Administration (FDA) for use in PET imaging of prostate cancer. Among them are ^68^Ga-PSMA-11, ^18^F-DCFPyL, and ^18^F-rhPSMA-7.3. They provide high detection rates and diagnostic accuracy compared to conventional imaging techniques [24,25]. Additionally, they can be utilized for target interrogation to determine patient eligibility for ^177^Lu-PSMA-617 (Pluvicto™) treatment and to evaluate responses to PSMA targeted therapy.

Monoclonal antibodies targeting PSMA can be radiolabeled with specific radionuclides that are chosen for particular tumor characteristics and clinical indications, whether it be for targeted radiodiagnosis or systemic radiotherapy, and may serve as an alternative or have a complementary role to PSMA small molecule radioligands.

The discovery of novel cell surface biomarkers and the development of mAbs with high specificity for these biomarkers could soon lead to an increased role for radiolabeled mAbs in prostate cancer. Accordingly, the accumulated experiences and lessons learned from previous mAb designs (failures and successes) are contributing to the development of the next generation of antibody-based cancer imaging and therapeutic agents.

## Figures and Tables

**Figure 1 cancers-15-04537-f001:**
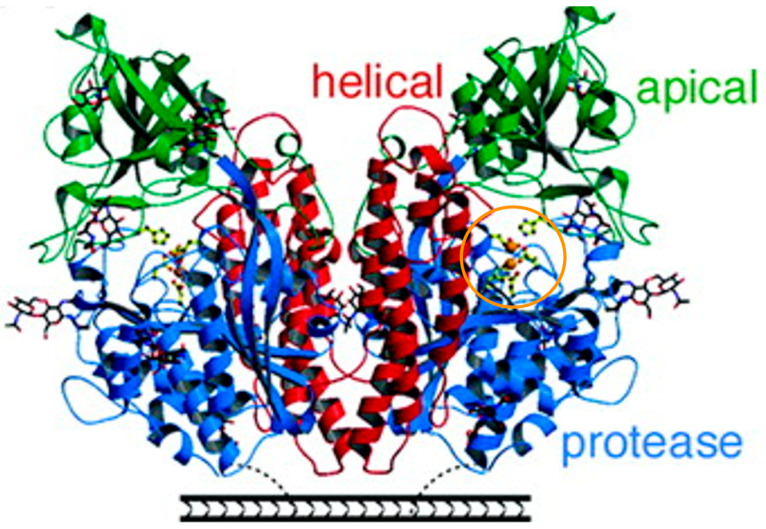
Ribbon diagrams of side and top view of PSMA. A surface rendering of PSMA in which the helical domain is red, the apical domain is green, the protease domain is blue, and zinc ions are depicted in orange circle. Adapted with permission from Davis MI et al. Crystal structure of prostate-specific membrane antigen, a tumor marker and peptidase [9].

**Figure 2 cancers-15-04537-f002:**
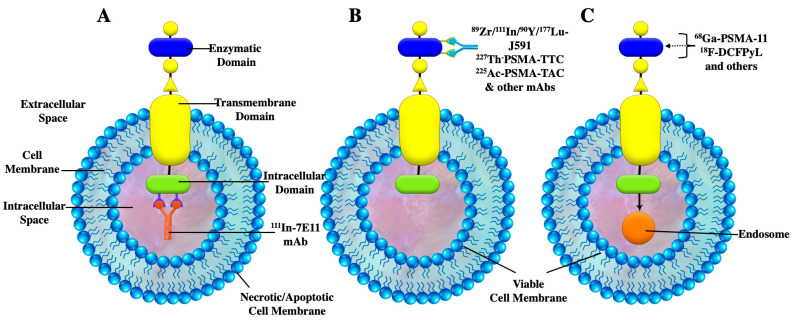
Diagrammatic representation of cellular transmembrane, mAbs targeting PSMA, and PSMA ligands: (**A**) Binding of 7E11 and other non-internalizing mAbs to the intracellular domain in a necrotic or apoptotic state. In this case, the internalization does not occur in viable cells; (**B**) Binding of J591, PSMA-TTC, and PSMA-TAC mAbs to the enzymatic domain. Rapid internalization occurs after binding to the receptor; (**C**) Binding of small molecules and ligands to the enzymatic domain. Intracellular transport is activated where endosomal activity results in cleavage and release of ligands into the cytoplasmic compartment.

**Figure 3 cancers-15-04537-f003:**
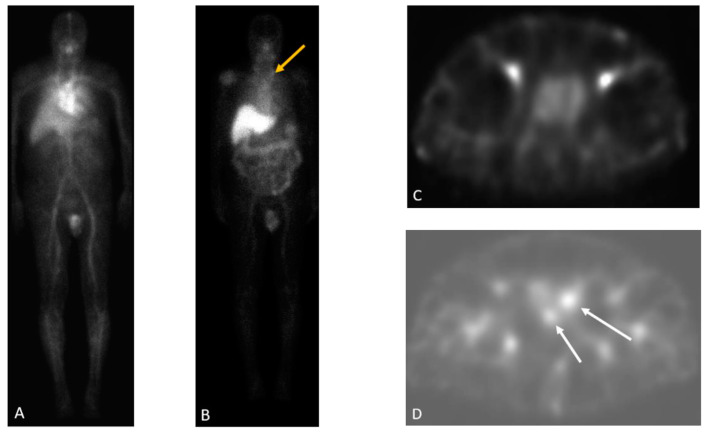
Sixty-four-year-old man with a history of radical prostatectomy for prostate cancer (Gleason 4 + 4) with rising PSA. ^111^In-capromab pendetide whole-body blood pool (**A**) and 96 h delayed (**B**) images demonstrate a focus of increasing radiotracer activity in the left neck base corresponding to a metastatic supraclavicular (Virchow) node (yellow arrow in (**B**)). Single-Photon Emission Computed Tomography (SPECT) blood pool (**C**) and delayed (**D**) images of the pelvis demonstrate two foci of increasing radiotracer activity in the right and left prostate bed, compatible with local recurrence (white arrows). With permission, this image was originally published by Jetty et. al. [50].

**Figure 5 cancers-15-04537-f005:**
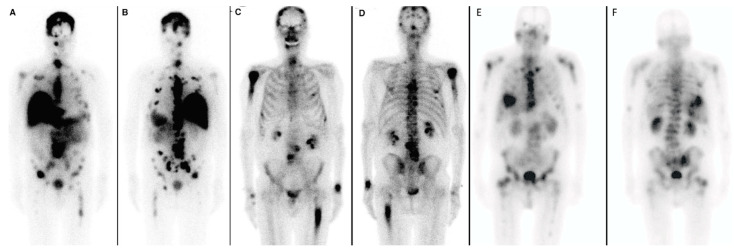
(**A**,**B**) An example of radiolabeled mAb ^177^Lu-J591 imaging. Anterior (**A**) and posterior (**B**) planar images obtained via dual head gamma camera of sites of uptake 7 days after ^177^Lu-J591 administration. No significant radiotracer uptake is seen in the salivary glands on the ^177^Lu-J591 images. (**C**,**D**) ^99m^Tc-MDP bone scan. Anterior (**C**) and posterior (**D**) planar images of pretreatment skeletal metastases are shown. (**A**–**D**) adapted with permission from Tagawa et al. [64]. Representative anterior (**E**) and posterior (**F**) planar images of a 71-year-old man with metastatic castration resistant prostate cancer obtained 4 h after the administration of 200 mCi of ^177^Lu-PSMA-617 demonstrate extensive osseous metastases involving the axial and appendicular skeleton and PSMA avid liver metastases. Lacrimal and salivary glands show insignificant uptake of ^177^Lu-PSMA-617.

**Figure 6 cancers-15-04537-f006:**
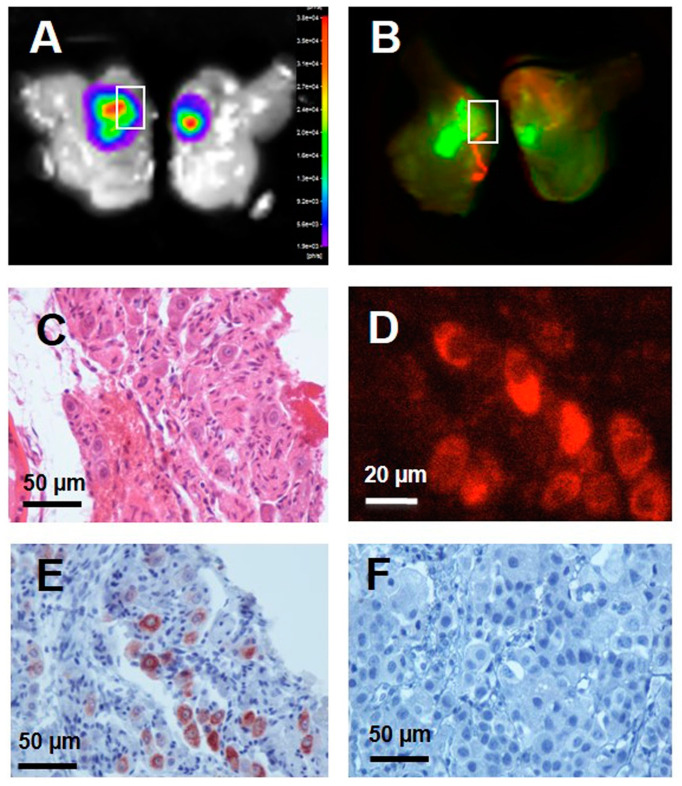
Detection of the X770-scFvD2B probe in the prostate 72 h after in vivo injection. On bisected prostates (**A**) BLI revealed a tumor and (**B**) fluorescence scanning revealed the X770-scFvD2B probe. On the paraffin prostate slice (4 μm) tumor cells were revealed (**C**) by hematoxylin-eosin-saffron staining and (**D**) X770-scFvD2B probe was detected by X770 fluorescence or (**E**) scFv His-tag detection using anti-His tag antibody (immunoperoxydase/DAB labeling). (**F**) No scFv His-tag was detected by immunohistochemistry after in vivo injection of the control fragment X770-scFvD2BGF7.7. (**A**–**F**) adapted with permission from Mazzocco, C. et al. [86].

**Figure 7 cancers-15-04537-f007:**
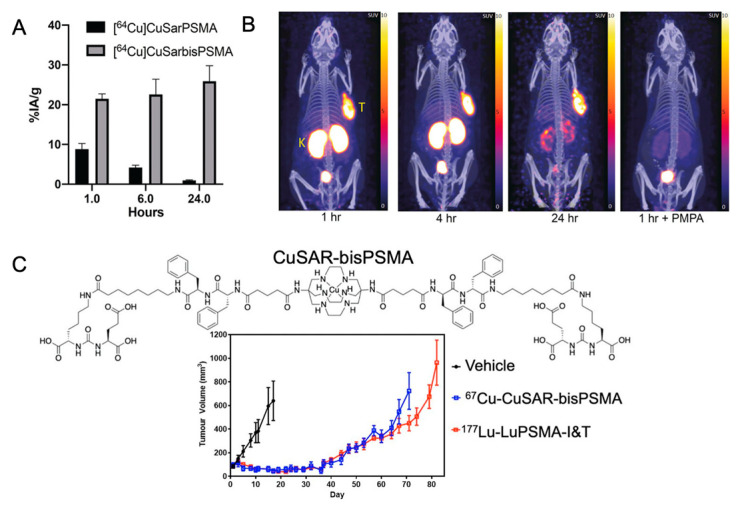
(**A**) *Ex vivo* tumor uptake expressed as percent injected activity per gram tissue (%IA/ g) (mean ± SEM, *n* = 3/group) in LNCap-tumor-bearing NSG mice following injection of either ^64^Cu-CuSarPSMA (2 MBq, 0.9 nmol of peptide) or ^64^Cu-CuSarbisPSMA (2 MBq, 0.2 nmol of peptide). (**B**) PET/CT images (maximum intensity projection, MIP) of LNCap-tumor-bearing NSG mice following injection of ^64^Cu-CuSarbisPSMA (2–3 MBq). Tumor (T) and kidney (K) uptake was blocked by co-administration of PSMA-specific inhibitor (PMPA) [27]. (**C**) Chemical structure of ^64/67^Cu-CuSarbisPSMA and inhibition of LNCaP tumor growth after treatment with either ^67^Cu-CuSarbisPSMA or ^177^Lu-LuPSMA I&T, expressed as mean tumor volume (±SEM) (n = 5) This research was originally published in JNM [92].

**Table 1 cancers-15-04537-t001:** Selected radionuclides used in clinical trials and labeled with monoclonal antibodies for treatment of prostate cancer.

Radioligand Radiolabeled mAbs	Half-Life	Emission α Particle	Max β+/− or Energy (MeV)	Max γ Particle Energy (MeV)	Clinical Use	ClinicalTrials.gov Identifier
^64^Cu-TLX592	12.7	AE, β+	0.653	1.346	Auger therapy, PET	NCT04726033
^64^Cu-CuSarbisPSMA						Pre-clinical only
^89^Zr-Df-IAb2M	78.4 h	β	0.909		PET	NCT02349022
^89^Zr-J591						NCT01543659
^111^In-7E11	8 d	AE, γ		0.171; 0.245	Auger therapy, SPECT	NCT02349022, NCT00992745
^177^Lu-J591 -	6.7 d	AE, β−, γ	0.177; 0.385; 0.5	0.208	Auger therapy, β therapy, SPECT	NCT00195039, NCT00538668
^177^Lu-DOTA-TLX591						NCT04786847, NCT04876651
^212^Pb-TCMC-YS5	10.6 h	β−			β therapy (daughter α)	NCT05245006
^225^Ac-J591	10.0 d	α	5.6-5.793		α therapy (daughter β, SPECT)	NCT03276572, NCT04506567, NCT04946370
^225^Ac-PSMA-TAC						NCT05219500
^225^Ac-DOTA-YS5						Pre-clinical only
^227^Th-PSMA-TTC	18.7 d	α	5.9		α therapy	NCT03724747
**Probe for fluorescence imaging**						
X770-scFvD2B					Fluorescent scFv of the anti-PSMA antibody	Pre-clinical only

β = beta therapy; α = alpha therapy.

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
