# Peer review of "Current Status of Radiolabeled Monoclonal Antibodies Targeting PSMA for Imaging and Therapy"

_cancers, 2023, doi:10.3390/cancers15184537_

Round 1

Reviewer 1 Report

This review provides an interesting analysis of imaging and radiotherapy of PSMA using antibodies (Ab). I have some suggestions to improve the paper:

It might be beneficial to either add a paragraph about small molecule tracers labeled with Ga-68 and F-18, which are more suitable for imaging PSMA in PCa than Ab-based tracers, or consider removing "imaging" from the title of the paper.

The organization of the manuscript could be improved. I suggest including a paragraph in the introduction section discussing the pros and cons of using Ab for radiotherapy (and imaging) before delving into specific Ab.

It would be helpful to provide more details on preclinical studies involving PSMA-dimer and albumin binding motif tracers. Specifically, focus on how these tracers have increased half-life in the blood and tumor uptake without necessarily affecting uptake in unwanted organs. Additionally, mention studies that have evaluated these tracers in PCa patients.

It would be advisable to rearrange the discussion regarding the use of J591 Ab for PET/Lu-177 and Ac-225 radiotherapy so that it is presented before moving on to another Ab. Additionally, it may be more suitable to compare J591 Ab with PSMA-617 in a different section of the manuscript. Currently, the placement of this section in the middle of the manuscript disrupts the flow of the content and makes it difficult to follow.

Clarify the difference between 177Lu-DOTA-rosopatamab and TLX591. In the company's advertisement, it was stated that "177Lu-DOTA-HuJ591-CHO (TLX591) is a radioimmunoconjugate comprised of the humanized IgG1 mAb rosopatamab, linked to the low energy beta-emitting radioisotope lutetium-177 (177Lu) via the bifunctional chelating agent DOTA-NHS ester." So, is this the same Ab?

In section 2.8, I noticed a sentence stating that "TLX591 is particularly effective at targeting tumor expressed PSMA and is cleared by the liver. Thus, it may be a more appropriate agent for targeting 225Ac, as it retains its specificity for tumor expressed PSMA and is cleared by the liver." I believe this sentence should be included in the TLX591 section, or at the very least, specify that TLX592 also undergoes metabolism in the liver, hence...

Lastly, the future perspective section could benefit from a more in-depth discussion on the usage of Ab-fragments. Your readers would appreciate more insights into this option.

Reviewer 2 Report

 Lucia Martiniova et al. comprehensively reviewed the “Current Status of Radiolabeled Monoclonal Antibodies Targeting PSMA for Imaging and Therapy”. The concept of highlighting the mAbs for PSMA Imaging and therapy with ongoing/past agents is encouraged.

The paper under evaluation is a well-written and comprehensive narrative/expert review focused on mAbs-based PSMA-targeted theranostic approaches in preclinical/clinical settings. I congratulate the authors for the excellent article. Minor comments are below,

1)    Page 3: Figure 1: Mabs mentioned instead of mAbs. Please correct it.

2)    Page 6: half of the Figure 4 description is in the wrong font size

3)    Page 7: Radiolabeling of J591 mAb with alpha-emitting payloads was investigated after the J591 mAb was successful in PC theranostic applications; Which applications? Provide references to the study in the sentence.

4)    I suggest adding more figures from the cited literature in the introduction and discussion sections.

5)    I suggest adding the period of collecting the literature for the review.

Reviewer 3 Report

The article by Martiniova et al. discusses the latest status of radiolabeled monoclonal antibodies which target PSMA for detecting/treating prostate cancer.

Some suggestions to improve this article are:

1. Figure legend for figure 2 require elaboration.

2. The conclusion section needs to be re-written with more details.

3. It seems section 2.4 and 2.9 are missing from the article.

Reviewer 4 Report

The prostate-specific membrane antigen (PSMA) gets overexpressed in prostate cancer cells and has emerged as a crucial target for its treatment. This is an interesting review where the authors described the radiolabeled monoclonal antibodies (mAbs) that can target PSMA. I would like to propose the following:

1. Please include the preclinical studies in the table.

2. Please make the introduction part slightly more comprehensive.

3. Please include the perspectives on radiolabeled monoclonal antibodies against PSMA in the conclusion; basically, the conclusion is missing this concept completely although the whole review is based on this concept.

The quality of English language seemed to be alright.

Round 2

Reviewer 3 Report

The revised version of the manuscript looks better.